# Clinical Predictors of Underlying Histologic Activity in Patients with Lupus Nephritis: A Focus on Urinary Soluble CD163

**DOI:** 10.3390/jcm14176162

**Published:** 2025-08-31

**Authors:** Bogdan Obrișcă, Alexandra Vrabie, Ștefan Lujinschi, Roxana Jurubiță, Valentin Mocanu, Andreea Berechet, Bogdan Sorohan, Andreea Andronesi, Gabriela Lupușoru, Camelia Achim, Georgia Micu, Dana Manda, Catalina Poalelungi, Nicu Caceaune, Simona Dima, Gener Ismail

**Affiliations:** 1Nephrology Department, “Carol Davila” University of Medicine and Pharmacy, 020021 Bucharest, Romania; alexandra.vornicu@drd.umfcd.ro (A.V.); stefan-nicolaie.lujinschi@drd.umfcd.ro (Ș.L.); roxana_jurubita@yahoo.com (R.J.); valentin-dumitrel.mocanu@drd.umfcd.ro (V.M.); bogdan.sorohan@umfcd.ro (B.S.); andreea.andronesi@umfcd.ro (A.A.); gabriela.lupusoru@umfcd.ro (G.L.); camelia.achim@umfcd.ro (C.A.); dima.simona@gmail.com (S.D.); gener.ismail@umfcd.ro (G.I.); 2Department of Nephrology, Fundeni Clinical Institute, 022328 Bucharest, Romania; andreea.berechet93@yahoo.com (A.B.); elenageorgia.micu@gmail.com (G.M.); nicu.caceaune@gmail.com (N.C.); 3Research Department, “C.I. Parhon” National Institute of Endocrinology, 011863 Bucharest, Romania; dana.manda@gmail.com (D.M.); catalina_1609@yahoo.com (C.P.); 4Center of Excellence in Translational Medicine, Fundeni Clinical Institute, 022328 Bucharest, Romania

**Keywords:** lupus nephritis, biomarker, histology, usCD163, activity index

## Abstract

**Background/Objectives**: We sought to evaluate the clinical predictors of underlying histologic activity in patients with lupus nephritis (LN), with a focus on urinary soluble protein CD163 (usCD163). **Methods**: We conducted a retrospective, cross-sectional study of forty-two consecutive LN patients with concurrent determination of usCD163 at the moment of kidney biopsy. A first morning void prior to the kidney biopsy was collected and usCD163 was measured by a commercial ELISA assay (EUROIMMUN, Lubeck, DE). **Results**: The study cohort had a median age at the moment of kidney biopsy of 33.5 (IQR: 24–42.7) years. The mean eGFR and median 24 h proteinuria were 76.6 ± 33.9 mL/min/1.73 m^2^ and 1.98 (IQR: 0.83–4.52) g/day. The median activity (AI) and chronicity (CI) indices were 7 (IQR: 3–11) and 3 (IQR: 1–5), respectively. usCD163 significantly correlated with 24 h proteinuria (*r* = 0.7, *p* < 0.001), hematuria (*r* = 0.51, *p* < 0.001), and serum complement levels, C3 (*r* = −0.5, *p* = 0.001) and C4 (*r* = −0.32, *p* = 0.03), but not with eGFR (*r* = −0.23, *p* = 0.14). Regarding the histological parameters, usCD163 significantly correlated with the AI and the individual active lesions (except for fibrinoid necrosis), but not with CI or any chronic lesion. usCD163 had a higher AUC compared to the classical measures of renal involvement (proteinuria, hematuria, eGFR) for discriminating an elevated AI, but the differences between AUC reached statistical significance only for hematuria. Thus, the AUC of usCD163 was 0.74 (95%CI, 0.58–0.86) for an AI over 2, an AUC of 0.77 (95%CI, 0.61–0.88) for an AI over 3 and an AUC of 0.74 (95%CI, 0.57–0.86) for an AI of at least 9. The optimal cutoff value for usCD163 identified for all AI thresholds evaluated was 296.2 ng/mmol. **Conclusions:** usCD163 correlates with glomerular inflammation, being able to discriminate histologic activity from chronicity in patients with LN and identify minimal histologic activity, although it did not significantly outperform proteinuria.

## 1. Introduction

Lupus nephritis (LN) is a common feature of systemic lupus erythematosus (SLE), and despite significant improvements in outcome over decades (with decreasing mortality and risk of end-stage kidney disease), there is still a substantial residual morbidity (from both disease-related and treatment-related factors) in these patients [1]. Despite the advent of several new disease-modifying therapies, the current complete renal response rates do not exceed 50% after two years of therapy, while a substantial proportion of patients (up to 20–30%) still progress to end-stage renal disease (ESRD) over years [2]. The failure of the LN outcomes to improve in parallel to the advances made in the understanding of its pathogenesis and pharmacotherapy resides on the lack of a consensual agreement regarding optimal risk stratifying algorithms to guide patient management [2]. Thus, recent efforts focused on defining optimal thresholds for proteinuria, renal function, or steroid dose at different time points to further incorporate them into clinical trials outcome measures [3].

To date, clinical and laboratory tests cannot reliably reflect histologic findings, and kidney biopsy remains the gold standard for diagnosis and adequate classification of LN [4]. Moreover, there is a significant lack of concordance between the clinical and histological features in LN. Thus, patients with low-level proteinuria (<0.5 g/day) may harbor a severe underlying histology and those that achieve a complete renal response may have significant residual histologic activity [2]. While the need for a *per-protocol* repeat kidney biopsy is gaining popularity and has been shown to be an invaluable tool for risk stratification and treatment guidance in LN, it remains an invasive procedure with potential serious complications [2]. Moreover, the use of kidney biopsy as a surveillance method may further be limited by a lack of agreement on the number of biopsies needed and optimal timing to adequately guide patient management [2].

Thus, there is an unmet need to identify potential biomarkers to reflect more accurately the underlying histologic activity than the current “classical” measures of LN activity (e.g., proteinuria) [2]. One of the most promising non-invasive tools to assess disease activity in LN is the measurement of urinary soluble CD163 (usCD163) [5]. CD163 is a transmembrane protein functioning as a scavenger receptor for the hemoglobin–haptoglobin complexes and is expressed mainly on activated M2c macrophages [5]. Macrophages play crucial roles in LN contributing to both the enhancement of inflammation and tissue repair [6]. Accordingly, in immune complex-mediated glomerular disorders (such as LN), anti-inflammatory M2c macrophages may be the predominant tissue infiltration phenotype, while in disease flares, there may be a transition towards proinflammatory macrophages [7]. The soluble CD163 is presumably shed in the urine by intrarenal M2c macrophages and has been proposed to be a promising biomarker of the underlying glomerular inflammation in LN, ANCA-associated vasculitis, or IgA Nephropathy [5,8,9]. Thus, the level of usCD163 has been shown to be higher in those with active LN compared to patients with inactive LN or SLE without renal involvement and to correlate with LN class or active lesions [10]. Nonetheless, whether usCD163 has a better capacity to reflect histologic activity in comparison to the classical measures of LN activity and the threshold for defining elevated levels remain largely unknown.

Thus, we sought to evaluate the performance of different clinical variables to reflect underlying histologic activity with a focus on defining the utility of usCD163 as a potential biomarker in LN.

## 2. Materials and Methods

### 2.1. Study Design and Population

This is a retrospective study that included a cross-sectional and a longitudinal evaluation of usCD163 in patients with LN. In the cross-sectional analysis, we included all consecutive patients (n = 42) with SLE and LN who underwent a kidney biopsy between January 2022 and December 2023 in our department and had a concurrent, previously stored urine sample. Patients underwent a kidney biopsy either for the initial diagnosis of LN (n = 28), for cause (flare biopsy, n = 2), or as part of a protocol biopsy (post-induction therapy biopsy, n = 2, or during maintenance therapy biopsy, n = 10). In the longitudinal analysis, a subset of the previous patients (n = 4) with multiple urine samples collected and stored during induction therapy along with a post-induction repeat kidney biopsy were included. The inclusion criteria were age over 18 years, histologic confirmation of LN, and a concurrent urine sample stored from the moment of kidney biopsy. We excluded patients with ages under 18 years whose renal biopsy specimens contained less than 8 scorable glomeruli, those with usCD163 assessment after the moment of the kidney biopsy, and those with other glomerular disorders (n = 7).

The clinical and laboratory variables were obtained by reviewing the patient’s medical records at the time of kidney biopsy and included age, gender, renal function [serum creatinine and estimated glomerular filtration rate (eGFR) according to the 2009 CKD-EPI equation] [11], albumin-to-creatinine ratio (mg/g) and 24 h proteinuria (g/day), urinary creatinine (mmol/L), hematuria (cells/µL), C3 and C4 levels (mg/dL), and serum antinuclear antibodies (ANA, U/mL). These patients received immunosuppressive therapy in accordance with the current KDIGO or EULAR treatment guidelines [12,13].

The study objective was to evaluate the correlation of usCD163 with clinical and histological variables. In addition, we sought to evaluate the ability of usCD163 to discriminate underlying severe or residual histologic activity as defined by different thresholds of the activity index.

This study was conducted with the provision of the Declaration of Helsinki and the protocol was approved by the local ethics committee (The Ethics Council of Fundeni Clinical; Registration number: 59389; 3 November 2022). All patients provided a signed informed consent form before study enrollment.

### 2.2. Kidney Biopsy and Urinary Soluble CD163 Assessment

Kidney biopsies slides were classified according to the International Society of Nephrology/Renal Pathology Society LN criteria and scored according to the modified National Institutes of Health activity and chronicity indices [14]. The pathologists were unaware of the biomarker assessment. Several thresholds for defining an elevated activity index (AI) were evaluated in relation to clinical variables: an activity index over 2 (as previously identified by De Rosa et al. to reflect residual histologic activity after immunosuppressive therapy with a subsequent risk of renal flare) [15], an activity index over 3 [as previously identified by Parodis et al. as a risk factor for a subsequent flare in repeat kidney biopsies and incorporated in the REBIOLUP trial as a threshold to intensify the immunosuppressive therapy at 12 months (REBIOLUP trial, ClinicalTrials.gov: NCT04449991)] [4,16], and an activity index ≥9 to define severe histologic activity [17].

This study included only patients with previously collected urine samples from the biopsy day from the first morning void (prior to the kidney biopsy) that were rapidly processed and adequately stored. Urine was centrifuged at 2000× *g* for 10 min at 4 °C. Supernatants were collected and frozen at −80 °C until measurement. usCD163 was measured by an in vitro diagnostic (IVD) ELISA assay (EUROIMMUN, Lubeck, Germany), initially expressed as ng/mL and then the values were normalized to urine creatinine and further expressed as ng per mmol of urine creatinine. The urine samples were processed in duplicate with an intra-assay and an inter-assay coefficient of variability of <5% and <10%, respectively, while the lower detection limit was 0.05 ng/mL. A concomitant assessment of urinary albumin and creatinine was undertaken from the same urine samples.

### 2.3. Statistical Analysis

Continuous variables were expressed as either mean (±standard deviation) or median (interquartile range: 25th–75th percentiles), according to the distribution of the variables, and categorical variables as percentages. Differences between groups were assessed in case of continuous variables by Mann–Whitney test. Spearman’s rank correlation tests were used to assess the relationship between clinical and pathological variables. Performance characteristics of clinical variables to discriminate patients with an elevated activity index were assessed by receiver operating characteristic curve (area under the curve, AUC). To assess the predictive ability of different variables to identify an elevated activity index we calculated sensitivity, specificity, positive predictive value (PPV), negative predictive value (NPV), positive likelihood ratio (LR+), negative likelihood ratio (LR-) and accuracy with the corresponding 95% confidence intervals (95%CI), and Youden Index. Based on the calculated Youden Index we defined the optimal cutoff values for variables to identify an elevated activity index. In order to evaluate whether combining clinical parameters will improve the performance characteristics, several models were developed by progressively combining the first five variables with the highest AUC for a given activity index threshold. In addition, univariate and multivariate logistic regression analysis were performed to identify predictors of an elevated usCD163.

In all analyses, *p* values are two-tailed and all *p* values less than 0.05 were considered statistically significant. Statistical analyses were performed using the SPSS program (SPSS version 26, Chicago, IL, USA), GraphPad Prism version 9.3.1 (1992–2021 GraphPad Software, LLC), and MedCalc Statistical Software version 22.021 (MedCalc Software Ltd., Ostend, Belgium).

## 3. Results

### 3.1. Study Population

Forty-nine consecutive patients with SLE and renal involvement who underwent a kidney biopsy during the study period and had an available urine sample stored for usCD163 determination were considered for study inclusion. Of these, 6 patients with LN that did not have a concomitant usCD163 determination at the moment of the kidney biopsy and 1 patient with a diagnosis of lupus podocytopathy were further excluded from the analysis, leaving a final cohort of 42 patients. In addition, four of these patients had multiple assessments during induction therapy along with a repeat post-induction kidney biopsy (as part of our internal protocol management of patients with LN).

The characteristics of the study population are depicted in Table 1. This cohort had a median age at the moment of kidney biopsy of 33.5 (IQR: 24–42.7) years, with a female-to-male ratio of 4.25:1 and a median time from SLE diagnosis of 4 years (IQR: 1–8), while the median time from induction therapy initiation to biopsy was 1 month (IQR: 0–11.5). The mean eGFR and median 24 h proteinuria were 76.6 ± 33.9 mL/min/1.73 m^2^ and 1.98 (IQR: 0.83–4.52) g/day, respectively. The median unadjusted usCD163 level was 4.88 ng/mL (IQR: 0.96–12.26), while the normalized usCD163 was 626.4 ng/mmol creatinine (IQR: 113.4–2384.1). In terms of treatment, cyclophosphamide-based regimens were used in 47.6% of patients during induction period, while 38.1% received mycophenolate mofetil. For maintenance treatment, mycophenolate mofetil (either alone or in combination with calcineurin inhibitors) was used in 66.6% of cases, while 26.2% received azathioprine (either alone or in combination with calcineurin inhibitors).

In terms of histology, most patients had either class III (n = 15, 34.9%) or class IV LN (n = 15, 34.9%), while the median activity and chronicity indices were 7 (IQR: 3–11) and 3 (IQR: 1–5), respectively (Table 2). In addition, the proportion of patients with class II (n = 4, 9.5%), class IV with V (n = 4, 9.5%), and pure class V (n = 4, 9.5%) was similar. Six patients (14.2%) showed tertiary lymphoid organs on kidney biopsy.

### 3.2. Cross-Sectional Analysis Evaluating the Relation of Urinary Soluble CD163 with Clinical and Pathologic Parameters

The level of usCD163 was higher in patients with class IV LN (with or without class V) compared to those with class II or pure class V [median usCD163: class II—161.7 ng/mmol (IQR: 29.4–552); class III—185.5 ng/mmol (IQR: 16.2–1347.9); class IV—970.9 ng/mmol (IQR: 378.3–4970.1); pure class V—280.3 ng/mmol (IQR: 72.5–1846.8); class IV with V—1952.6 ng/mmol (IQR: 488.7–3058)] (Figure 1). The patient that had a diagnosis of lupus podocytopathy had a usCD163 level of 512.1 ng/mmol and a 24 h proteinuria of 6.5 g/day. Given that this patient would potentially represent a false positive case of elevated usCD163 in relation to the absence of glomerular inflammation, we further evaluated the relation between usCD163 and underlying histology with the exclusion of this patient from the analysis.

The level of usCD163 significantly correlated with 24 h proteinuria (*r* = 0.7, *p* < 0.001), albumin/creatinine ratio (*r* = 0.87, *p* < 0.001), hematuria (*r* = 0.51, *p* < 0.001), and serum complement levels, C3 (*r* = −0.5, *p* = 0.001) and C4 (*r* = −0.32, *p* = 0.03), but not with eGFR (*r* = −0.23, *p* = 0.14) (Appendix A). In terms of the histological parameters, usCD163 significantly correlated with the activity index and the individual active lesions (except for fibrinoid necrosis) but not with the chronicity index or any chronic lesions (Figure 2 and Figure 3). By comparison, the classical measures of LN activity (hematuria, proteinuria, or eGFR) showed weaker correlations with the activity index or its individual components with hematuria being significantly correlated only with endocapillary hypercellularity (*r* = 0.33, *p* = 0.03) and neutrophil/karyorrhexis (*r* = 0.32, *p* = 0.04), proteinuria with endocapillary hypercellularity (*r* = 0.41, *p* = 0.006) and cellular/fibrocellular crescents (*r* = 0.31, *p* = 0.04), and eGFR with interstitial inflammation (*r* = −0.42, *p* = 0.005) (Appendix A). 

In addition, the level of usCD163 correlated with activity index in all proteinuria subgroups and irrespective of chronicity index (Figure 4). As an example, one patient with a 24 h proteinuria of 0.44 g/day had an activity index of 14 and an elevated usCD163 (616.4 ng/mmol). Contrarily, another patient with class V lupus nephritis, a 24 h proteinuria of 2.9 g/day, and an activity and chronicity index of 0 had a low usCD163 level (263.3 ng/mmol). Similarly, usCD163 can differentiate an underlying histologic activity in patients with the same level of proteinuria. Thus, one patient had a proteinuria of 1.58 g/day, activity index of 0, and a low usCD163 of 296.2 ng/mmol, while another patient with a similar proteinuria (1.65 g/day) but an activity index of 11 had an elevated usCD163 of 670.7 ng/mmol. Moreover, we did not identify any difference in terms of usCD163 level according to the presence of tertiary lymphoid organs (Figure 5).

### 3.3. Cross-Sectional Analysis Evaluating the Performance Characteristics of Clinical Variables to Identify a High Activity Index

We further evaluated the performance characteristics of clinical variables to identify patients with an elevated activity index (Table 3 and Table 4, Figure 6). usCD163 had a higher AUC compared to the classical measures of renal involvement (proteinuria, hematuria, eGFR) for all thresholds for defining an elevated AI, with an AUC of 0.74 (95%CI, 0.58–0.86) for an AI over 2, an AUC of 0.77 (95%CI, 0.61–0.88) for an AI over 3, and an AUC of 0.74 (95%CI, 0.57–0.86) for an AI of at least 9 (Table 3). When evaluating the differences between areas, usCD163 had an AUC significantly higher only when compared to hematuria (difference between areas, 0.19 (95%CI, −0.003 to 0.39), *p* = 0.02) and serum C4 (difference between areas, 0.25 (95%CI, 0.01 to 0.49), *p* = 0.03) at an AI over 3 and compared to serum C4 (difference between areas, 0.22 (95%CI, 0.02 to 0.42), *p* = 0.02) at an AI of at least 9, but not for the rest of the variables (Table 3). Nonetheless, usCD163 did not outperform 24 h proteinuria. The optimal cutoff value for usCD163 identified for all AI thresholds evaluated was 296.2 ng/mmol. Thus, an usCD163 > 296.2 ng/mmol had a positive predictive value of 92.3% (95%CI, 77.6–97.6) for an AI over 2 and of 92.3% (95%CI, 77.3–97.6) for an AI over 3 (Table 4). Progressively incorporating other clinical or immunological variables into models to identify minimal histologic activity (AI over 2 or 3) further increased the specificity and positive predictive value to 100%, albeit at a progressively inferior sensitivity and negative predictive value and accuracy, while the differences between AUC were not statistically significant when compared to usCD163 alone (Table 3 and Table 4, Figure 7). When evaluating the capacity of different variables to discriminate patients with severe glomerular inflammation (AI of at least 9), usCD163 had a similar diagnostic accuracy with hematuria and C3 level, but with better sensitivity [88.8% (95%CI, 65.2–98.6)] and a negative predictive value [87.5% (95%CI, 64.4–96.4)] (Table 4, Figure 6).

The cutoff for usCD163 defined by receiver operating characteristic curve analysis was further incorporated into logistic regression analysis as a dependent variable in order to identify its independent predictors (Appendix A). After multivariate adjustment, the activity index (OR, 1.3 for 1 point; 95%CI, 1.05–1.59) and proteinuria (OR, 1.9 for 1 g/day; 95%CI, 1.06–3.39) were identified as independent predictors of elevated usCD163 (>296.2 ng/mmol).

### 3.4. Longitudinal Analysis of usCD163 During Induction Therapy

Four patients underwent multiple assessments of usCD163 during the induction therapy consisting of either steroids and cyclophosphamide (Euro-Lupus regimen, three patients) or mycophenolate mofetil and rituximab (one patient) along with a post-induction repeat kidney biopsy. The repeat kidney biopsy was undertaken at a mean 7.05 ± 0.46 months following the diagnostic kidney biopsy. During induction therapy, the usCD163 level gradually decreased from a median 524.5 ng/mmol (IQR: 335.2–1238.6) at the baseline biopsy to a median of 6.42 ng/mmol (IQR: 0.75–15.1) at the post-induction kidney biopsy. This was paralleled by a concomitant decrease of proteinuria [from a median of 1.57 g/day [(IQR: 1.3–1.66) to 0.09 g/day (IQR: 0.03–0.17)] and of the activity index [from a median of 11.5 (IQR: 11–13.7) to 7.5 (IQR: 6.2–8.7)] (Figure 8). In addition, the chronicity index remained stable [median, 4 (IQR: 2.5–6.2) and 5 (IQR: 4.7–5.5)]. At the end of the induction therapy, all patients had a usCD163 below 296.2 ng/mmol and a proteinuria below 0.5 g/day.

## 4. Discussion

In this study, we showed that usCD163 correlates better with glomerular inflammation by comparison to the “classical” measures of LN activity. In addition, usCD163 was able to discriminate patients with minimal histologic activity with good performance characteristics, albeit not statistically significant compared to proteinuria. Moreover, a low usCD163 level was able to rule out severe glomerular inflammation with a high negative predictive value.

The assessment of renal function and proteinuria are currently the backbone of treatment indication and response criteria, but the clinical distinction between histologic activity and chronicity in LN remains challenging [2]. Nonetheless, recent data outline that a low-level proteinuria does not exclude significant histologic activity. In a recent study that enrolled 46 patients with SLE and proteinuria less than 0.5 g/day who underwent a kidney biopsy, it was shown that approximately 85% of cases had proliferative LN with a median activity index of 6 (and up to 14) [18]. Similarly, in our cohort, seven patients had a proteinuria at biopsy below 0.5 g/day with an activity index ranging from 0 to 14. These findings question the proteinuria level of >0.5 g/day proposed by the current clinical guidelines as a threshold for the indication of kidney biopsy in LN [13,19]. Furthermore, several repeat kidney biopsy studies highlight the inability of current measures of LN activity (proteinuria and renal function) to adequately assess the immunological response and to reflect the underlying histology [2,20]. Malvar et al. showed that, in a cohort of 69 patients with LN who underwent a post-induction repeat kidney biopsy (at 6 months), 29% of those who achieved complete clinical remission had a high persistent histologic activity at biopsy 2 (with an AI ≥ 5) and 62% of those with complete histologic remission (AI of 0) had a persistent proteinuria ≥0.5 g/day [21].

Accordingly, there is an unmet need to validate non-invasive biomarkers that correlate with histologic activity and to incorporate them into disease monitoring algorithms. Although several biomarkers have been evaluated in LN over the past decades, none have been translated from bench to bedside [22]. Nonetheless, urinary soluble CD163 emerges as a promising non-invasive biomarker that could possibly reflect better the underlying histology in LN [2].

Previous studies have shown that usCD163 is able to discriminate the presence of active LN from inactive LN/non-renal SLE or proliferative LN (class III and IV) from class II and V with an AUC ranging from 0.76 to 0.94 [10,23]. By comparison to previous reports, we have undertaken a different approach by assessing the biomarker capacity to discriminate different grades of histologic activity (e.g., minimal or severe histologic activity). As such, we selected an AI cutoff of 2 and 3 to define minimal histologic activity. Thus, De Rosa et al. showed that all patients with residual histologic activity (AI over 2) after 36 months of immunosuppressive therapy with at least 12 months of complete clinical remission relapsed during follow-up [15]. Moreover, the REBIOLUP trial set an AI threshold of 3 at the 12-month repeat kidney biopsy as an indication to intensify immunosuppressive therapy (REBIOLUP trial, ClinicalTrials.gov: NCT04449991) [16]. In addition, although an activity index cutoff to characterize severe LN has not been adequately defined, we used a threshold of 9 to reflect severe glomerular inflammation based on the results of the NIH trial [17]. First, our results are consistent with the previous reports regarding the strength of correlation with the activity index (*r* = 0.47 in our study vs. *r* = 0.4 in the study by Zhang et al. vs. *r* = 0.48 to 0.59 in the study by Mejia-Vilet et al.) and the lack of correlation with chronicity index [5,23]. In addition, usCD163 had a higher AUC compared to the classical measures of renal involvement (proteinuria, hematuria, eGFR) for all thresholds employed for defining an elevated AI, ranging from 0.74 to 0.77. At a cutoff of 296.2 ng/mmol, usCD163 had a good capacity to detect minimal histologic activity (AI over 2 or 3) with a positive predictive value of approximately 92% and a better accuracy than proteinuria, hematuria, or renal function. By contrast, a low usCD163 level was able to rule out severe glomerular inflammation (AI of at least 9) with a negative predictive value of 87.5%. Nonetheless, the statistical significance when comparing the differences between areas was reached only for hematuria and serum C4 but not for proteinuria. The reason for this might be the tight relation between usCD163 and glomerular permeability as expressed by proteinuria. While previous studies have shown that proteinuria accounted for approximately 40–50% of the variability in usCD163 levels, in our study, proteinuria or albumin/creatinine ratio were highly correlated with usCD163 levels (*r* = 0.7 and *r* = 0.87) [5]. Further insights come from our serial assessments. In our patients who underwent a post-induction repeat kidney biopsy, serial assessment showed a complete normalization of usCD163 levels that paralleled, but slightly preceded, the decrease in proteinuria, while the activity index decreased from 11.5 (IQR: 11–13.7) to 7.5 (IQR: 6.2–8.7), further highlighting a significant persistent residual activity. Thus, the reason for this might be related to the concomitant remission of proteinuria in these patients. Whether the urinary losses of usCD163 rather reflect the increased glomerular permeability, and thus proteinuria, remains to be answered. In previous studies assessing the utility of usCD163 as a biomarker in LN, usCD163 level was normalized to urinary creatinine, and whether a normalization to urinary albumin is better remains to be further evaluated [5,9,10,23].

Moreover, hypothesis has emerged regarding why a marker of an anti-inflammatory phenotype of macrophages correlates with glomerular inflammation [5]. M2c macrophages infiltrate the tissue during the healing phases and might reflect the capacity of the renal tissue to resolve the acute inflammation without significant progression of fibrosis [7]. In support of this hypothesis, Li et al. showed that in patients with IgA Nephropathy, steroid therapy was associated with a greater clinical response in those with higher usCD163 levels, leading to a higher decline in usCD163 level compared to placebo, and this translated into better long-term renal survival [24].

Apart from this aspect, the different rate of resolution of individual active lesions must be considered. Accordingly, it was shown that glomerular crescents, karyorrhexis, and fibrinoid necrosis usually resolve in several months, while the resolution of endocapillary hypercellularity, subendothelial hyaline deposits, and interstitial inflammation takes several years [25]. Thus, the impact of the severity and trend of each individual active lesion on the level of usCD163 must be considered. Given this consideration, monitoring usCD163 levels every 2–3 months during induction therapy would be suitable for optimally identifying the time for transitioning to maintenance therapy. In addition, in patients that are in complete clinical remission periodic assessment of usCD163 might be useful to identify early relapses.

Our study has several limitations that need to be acknowledged. First, this is a cross-sectional, single-center study that enrolled a limited number of patients (n = 42). The longitudinal analysis included only a few of these patients (n = 4) who had serial biomarker assessment during induction therapy to evaluate the trends of usCD163 in parallel to proteinuria and activity index. Accordingly, increasing the sample size by multicenter collaboration is needed to fully validate these findings. Thus, additional studies are needed to further validate the utility of usCD163 as a biomarker in LN and to evaluate the optimal assessment and normalization to other urinary variables. Accordingly, given the high correlation of usCD163 with proteinuria in our study, it remains difficult to distinguish whether an elevated level is due to increased glomerular permeability or due to active glomerular inflammation. Second, this is a retrospective study that included only patients with previously available urine samples that were processed and stored adequately for possible future assessments of different biomarkers. Thus, a prospective cohort with adequate longitudinal assessment is therefore needed to evaluate the role of usCD163 in tailoring the disease monitoring in lupus nephritis. Third, this study does not include a long-term follow-up data in all patients to evaluate whether a baseline usCD163 predicts a future renal response in a multivariate analysis. Nonetheless, our study is different from previous reports in that we evaluated the relation of usCD163 with different thresholds of AI that would reflect several clinical scenarios for immunosuppressive treatment adjustments (e.g., intensifying treatment for severe glomerular inflammation, decrease, or even withdrawal of therapy in cases with minimal or absent glomerular inflammation), although our cross-sectional study design would be indirect proof of this hypothesis. In addition, usCD163 is one of the potential biomarkers in LN that have validated in independent cohorts, deeming it a potential “liquid biopsy” [26]. However, given the heterogenous immune and pathological landscape of lupus nephritis, relying on a single urinary biomarker may provide a suboptimal assessment. Thus, as recently shown by Fava et al., a combination of urine biomarkers of intrarenal immunological activity (e.g., biomarkers of monocyte/neutrophil degranulation, macrophage activation, wound healing/matrix degradation, IL-16) would lead to a better characterization of proliferative lupus nephritis and prediction of clinical response [27]. Accordingly, as previously mentioned, the available urine samples could be further used to integrate these findings in the context of other putative urine biomarkers. Nonetheless, by comparison to other molecules, usCD163 possesses some additional qualities such as stability at room temperature for at least 1 week and for several years at −20 °C, while the ELISA method of measurement is easy to implement in clinical practice [9]. These aspects minimize the influence of the process of sample handling on the final assay measurement.

## 5. Conclusions

In conclusion, we have shown that usCD163 correlates with glomerular inflammation, being able to discriminate histologic activity from chronicity in patients with LN and identify minimal histologic activity, although it did not significantly outperform proteinuria. In addition, a low usCD163 level was able to rule out severe glomerular inflammation with a good negative predictive value. Thus, usCD163 could be a promising non-invasive biomarker for glomerular inflammation, albeit further studies are needed to define its role in risk stratification in these patients.

## Figures and Tables

**Figure 1 jcm-14-06162-f001:**
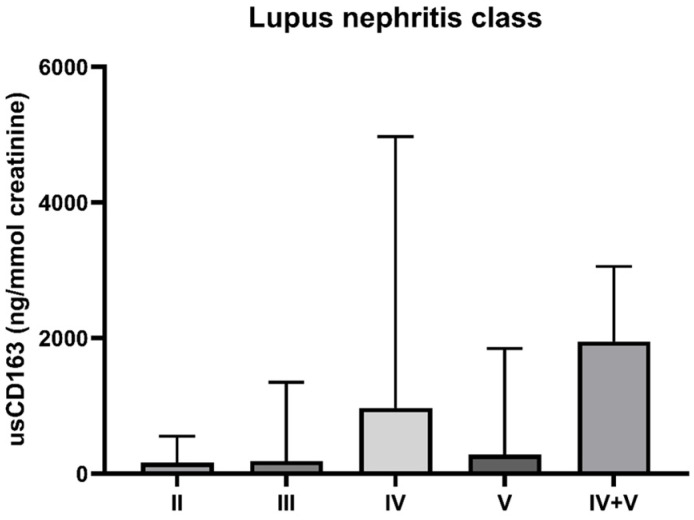
Level of usCD163 according to LN class.

**Figure 2 jcm-14-06162-f002:**
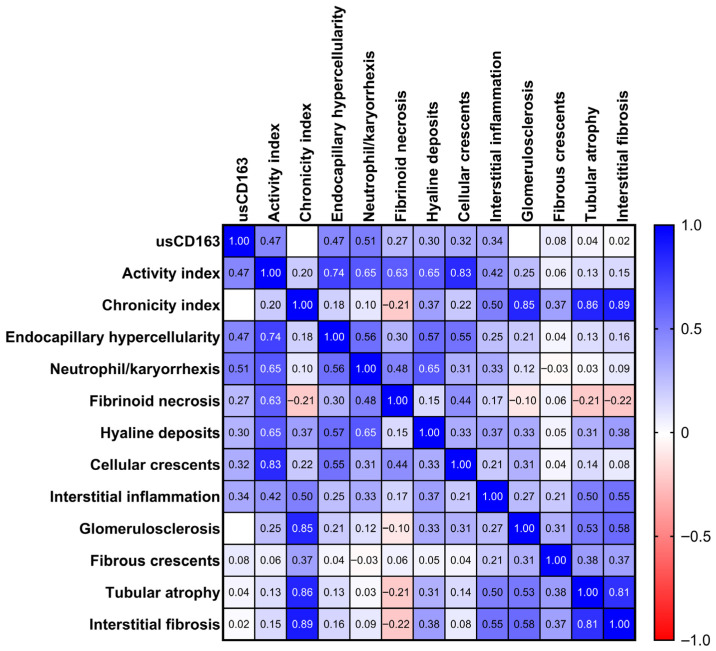
Correlation matrix of usCD163 with individual pathologic lesion.

**Figure 3 jcm-14-06162-f003:**
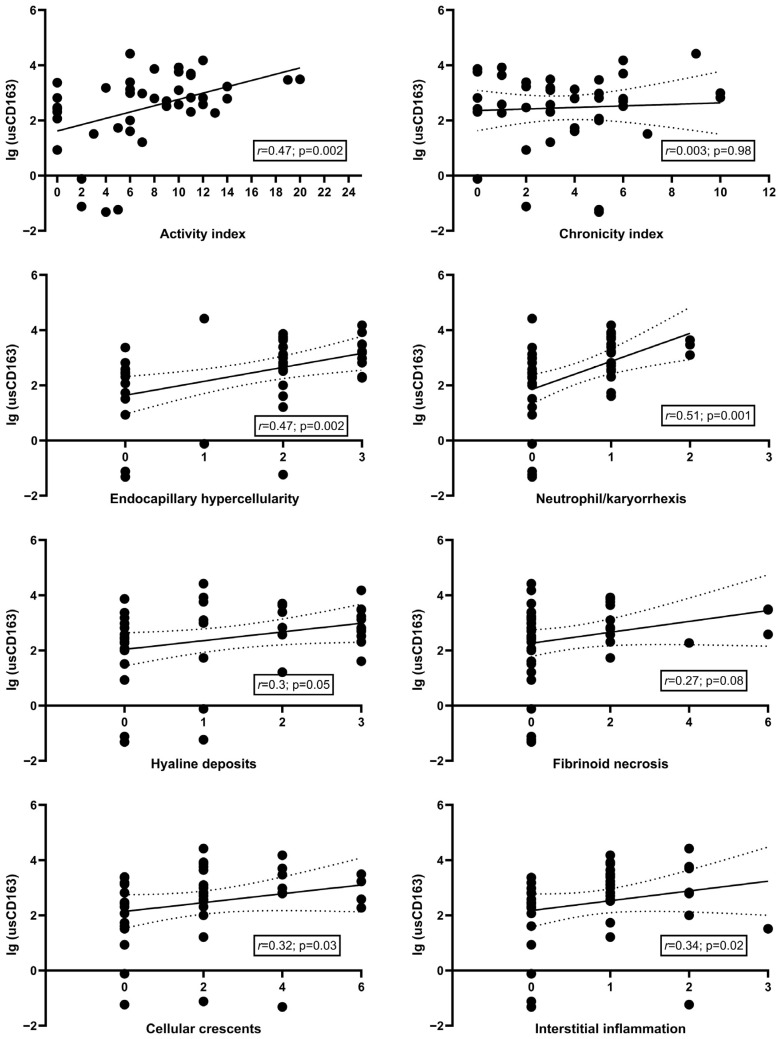
Level of usCD163 according to the severity of individual pathologic lesions (with simple linear regression and corresponding 95% confidence interval).

**Figure 4 jcm-14-06162-f004:**
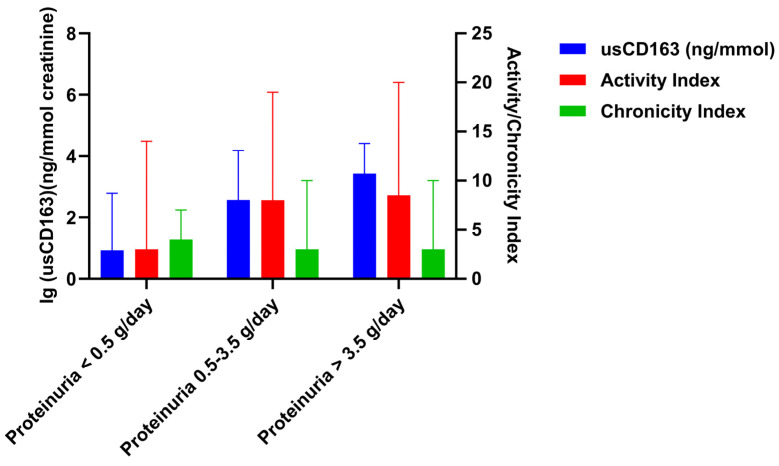
Level of usCD163, severity of activity, and chronicity index stratified by 24 h proteinuria.

**Figure 5 jcm-14-06162-f005:**
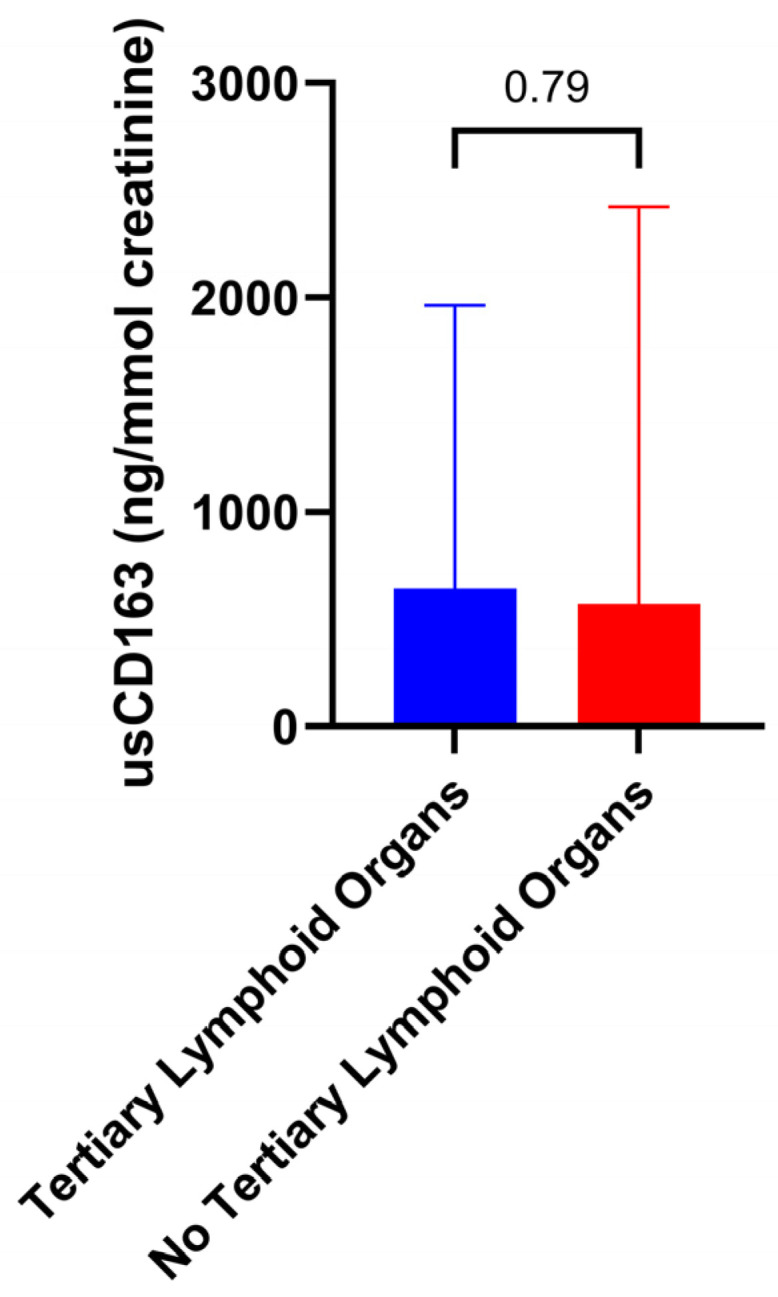
Level of usCD163 according to the presence of tertiary lymphoid organs.

**Figure 6 jcm-14-06162-f006:**
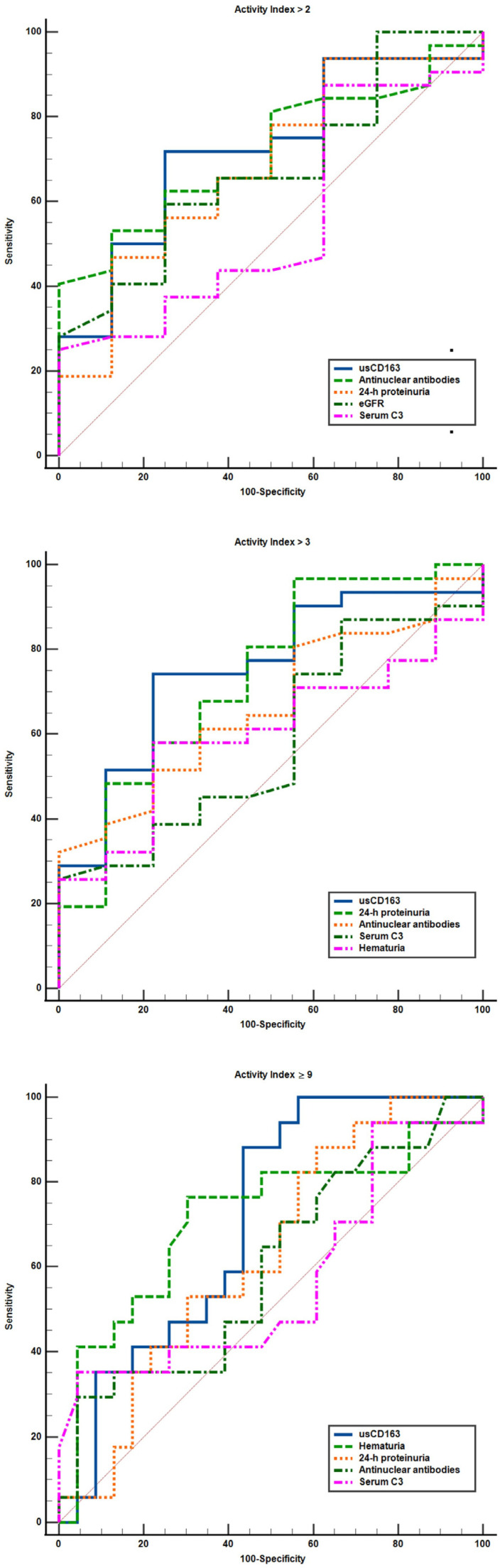
Performance characteristics of clinical variables to identify an elevated activity index.

**Figure 7 jcm-14-06162-f007:**
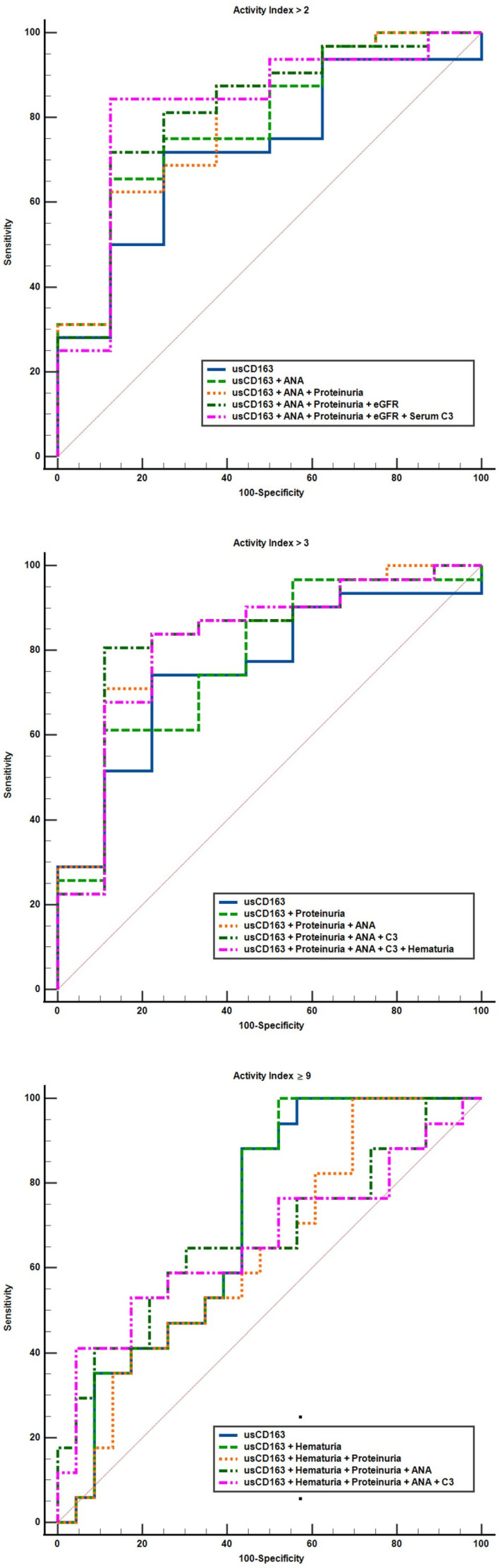
Performance characteristics of clinical variables to identify an elevated activity index (models incorporating usCD163 with other clinical or immunological variables).

**Figure 8 jcm-14-06162-f008:**
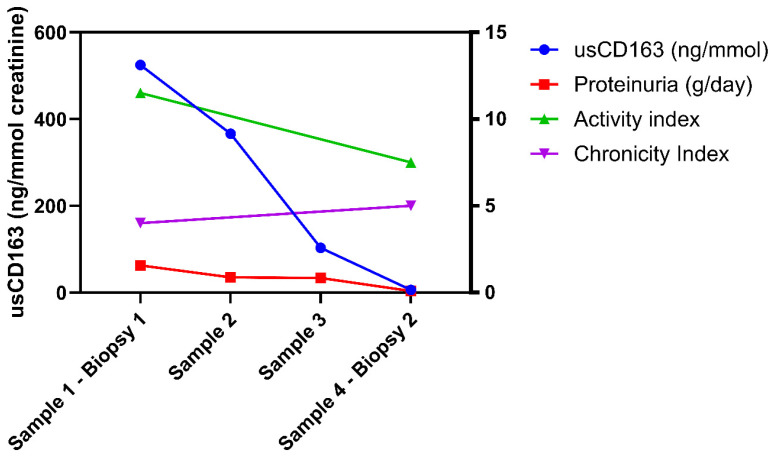
Longitudinal assessment of usCD163 during induction therapy.

**Table 1 jcm-14-06162-t001:** Baseline characteristics of the study cohort.

Clinical Variable	Value
**Age (y)**	33.5 (24–42.75)
**Gender (n, %F/%M)**	34/8 (81%/19%)
**Time since SLE diagnosis (years)**	4 (1–8)
**Time from induction therapy initiation to biopsy (months)**	1 (0–11.5)
**Serum creatinine (mg/dL)**	1.07 (0.8–1.34)
**eGFR (mL/min/1.73 m^2^)**	76.6 ± 33.9
**24-h proteinuria (g/day)**	1.98 (0.83–4.52)
**24-h proteinuria (n, % of pts)**	
<0.5 g/day	7 (16.7%)
0.5–3.5 g/day	21 (50%)
>3.5 g/day	14 (33.3%)
**Albumin/creatinine ratio (mg/g)**	849 (298–3770)
**Albuminuria (mg/dL)**	791 (230–3233)
**Urinary creatinine (mmol/L)**	8.4 (5.4–14.8)
**Hematuria (cells/µL)**	25 (3–54)
**Total cholesterol (mg/dL)**	214 (170–258)
**Triglycerides (mg/dL)**	139 (91–190)
**ANA (U/mL)**	5.8 (0.9–7.8)
**Anti-dsDNA antibody positivity (% of pts)**	77.5%
**Anti-Sm antibody positivity (% of pts)**	37.5%
**C3 (mg/dL)**	79.6 ± 32.9
**C4 (mg/dL)**	12.7 (7–20)
**Raw usCD163 (ng/mL)**	4.88 (0.96–12.26)
**usCD163 normalized (ng/mmol creatinine)**	626.4 (113.4–2384.1)
**Immunosuppressive treatment characteristics**	
**Use of steroids (% of pts)**	100%
** *Induction treatment (% of pts)* **	
Cyclophosphamide	47.6%
Mycophenolate mofetil	38.1%
Mycophenolate mofetil + calcineurin inhibitors	4.8%
Azathioprine	7.1%
**Use of rituximab during induction**	33.3%
** *Maintenance treatment (% of pts)* **	
Mycophenolate mofetil	59.5%
Azathioprine	23.8%
Mycophenolate mofetil + calcineurin inhibitors	7.1%
Azathioprine + calcineurin inhibitors	2.4%
Rituximab	2.4%

**Abbreviations:** F: females; M: males; eGFR: estimated glomerular filtration rate; pts: patients; ANA: antinuclear antibodies.

**Table 2 jcm-14-06162-t002:** Biopsy features of the study cohort.

Histological Variable	Value
**LN class [n, %]**	
II	4 (9.5%)
III	15 (35.7%)
IV without V	15 (35.7%)
Pure V	4 (9.5%)
VI	0%
IV with V	4 (9.5%)
**Activity Index**	7 (3–11)
Endocapillary hypercellularity	2 (0–2)
Neutrophils/karyorrhexis	1 (0–1)
Fibrinoid necrosis	0 (0–2)
Hyaline deposits	1 (0–3)
Cellular/fibrocellular crescents	2 (0–4)
Interstitial inflammation	1 (0–1)
**Chronicity Index**	3 (1–5)
Total glomerulosclerosis score	1 (0–2)
Fibrous crescents	0 (0–0)
Tubular atrophy	1 (0–1)
Interstitial fibrosis	1 (0–2)

**Abbreviations:** LN: lupus nephritis.

**Table 3 jcm-14-06162-t003:** Performance characteristics of clinical variables to identify an elevated activity index.

Variable	AUC	*p*	Cutoff	Youden Index	Differences of AUC (Versus 1)	*p*
**(I) Activity index over 2**						
(1) usCD163 (ng/mmol)	0.74 (0.58–0.86)	0.007	>296.2	0.51	-	-
(2) Hematuria (cells/µL)	0.54 (0.38–0.7)	0.65	>19.5	0.24	0.19 (−0.01 to 0.4)	0.07
(3) Proteinuria (g/d)	0.69 (0.53–0.83)	0.04	> 2.9	0.37	0.04 (−0.06 to 0.15)	0.46
(4) eGFR (mL/min)	0.62 (0.46–0.77)	0.25	≤81	0.27	0.11 (−0.16 to 0.39)	0.42
(5) C3 level (mg/dL)	0.57 (0.41–0.72)	0.47	≤48	0.24	0.16 (−0.08 to 0.41)	0.19
(6) C4 level (mg/dL)	0.52 (0.36–0.67)	0.82	≤8	0.25	0.22 (−0.02 to 0.46)	0.07
(7) ANA level (U/mL)	0.72 (0.55–0.84)	0.01	5.9	0.41	0.005 (−0.21 to 0.22)	0.95
1 + 7	0.79 (0.63–0.9)	0.001	-	0.53	0.06 (−0.07 to 0.21)	0.35
1 + 7 + 3	0.80 (0.64–0.91)	0.001	-	0.5	0.07 (−0.07 to 0.22)	0.32
1 + 7 + 3 + 4	0.81 (0.66–0.92)	<0.001	-	0.59	0.09 (−0.04 to 0.23)	0.17
1 + 7 + 3 + 4 + 5	0.82 (0.67–0.93)	<0.001	-	0.72	0.1 (−0.04 to 0.25)	0.17
**(II) Activity index over 3**						
(1) usCD163 (ng/mmol)	0.77 (0.61–0.88)	0.001	>296.2	0.55	-	-
(2) Hematuria (cells/µL)	0.57 (0.41–0.72)	0.45	>19.5	0.29	0.19 (−0.003 to 0.39)	0.05
(3) Proteinuria (g/d)	0.75 (0.58–0.86)	0.008	>2.9	0.4	0.02 (−0.08 to 0.12)	0.73
(4) eGFR (mL/min)	0.56 (0.4–0.71)	0.56	≤130	0.2	0.2 (−0.08 to 0.48)	0.16
(5) C3 level (mg/dL)	0.6 (0.44–0.75)	0.31	≤48	0.25	0.16 (−0.05 to 0.39)	0.15
(6) C4 level (mg/dL)	0.51 (0.35–0.66)	0.9	≤8	0.27	0.25 (0.01 to 0.49)	0.03
(7) ANA level (U/mL)	0.66 (0.5–0.81)	0.07	>7.6	0.32	0.09 (−0.15 to 0.32)	0.48
1 + 3	0.79 (0.63–0.9)	<0.001	-	0.52	0.02 (−0.04 to 0.09)	0.49
1 + 3 + 7	0.83 (0.67–0.93)	<0.001	-	0.62	0.07 (−0.05 to 0.2)	0.24
1 + 3 + 7 + 5	0.83 (0.67–0.93)	<0.001	-	0.69	0.07 (−0.06 to 0.21)	0.29
1 + 3 + 7 + 5 + 2	0.82 (0.66–0.92)	<0.001	-	0.62	0.06 (−0.07 to 0.2)	0.35
**(III) Activity index of at least 9**						
(1) usCD163 (ng/mmol)	0.74 (0.57–0.86)	0.002	>296.2	0.47	-	-
(2) Hematuria (cells/µL)	0.7 (0.54–0.83)	0.02	> 19.5	0.44	0.03 (−0.14 to 0.21)	0.71
(3) Proteinuria (g/d)	0.64 (0.48–0.78)	0.1	> 1.4	0.29	0.09 (−0.02 to 0.22)	0.13
(4) eGFR (mL/min)	0.51 (0.34–0.66)	0.95	≤100	0.22	0.23 (−0.03 to 0.49)	0.08
(5) C3 level (mg/dL)	0.58 (0.42–0.73)	0.37	≤41.2	0.29	0.15 (−0.03 to 0.34)	0.11
(6) C4 level (mg/dL)	0.51 (0.35–0.67)	0.87	≤4.6	0.11	0.22 (0.02 to 0.42)	0.02
(7) ANA level (U/mL)	0.6 (0.43–0.75)	0.28	>9.6	0.25	0.11 (−0.14 to 0.36)	0.39
1 + 2	0.74 (0.58–0.86)	0.002	-	0.5	0.002 (−0.006 to 0.01)	0.62
1 + 2 + 3	0.65 (0.49–0.79)	0.06	-	0.29	0.07 (−0.05 to 0.21)	0.24
1 + 2 + 3 + 7	0.67 (0.5 to 0.81)	0.06	-	0.34	0.04 (−0.18 to 0.26)	0.72
1 + 2 + 3 + 7 + 5	0.66 (0.49–0.81)	0.07	-	0.36	0.04 (−0.17 to 0.27)	0.68

**Abbreviations:** eGFR: estimated glomerular filtration rate; ANA: antinuclear antibodies.

**Table 4 jcm-14-06162-t004:** The accuracy of clinical variables at calculated cutoffs for identifying an elevated activity index (with 95% confidence interval).

Variable	Sensitivity	Specificity	PPV	NPV	LR+	LR-	Accuracy
**(I) AI over 2**							
(1) usCD163 > 296.2 ng/mmol	72.7% (54.4–86.7)	77.7% (39.9–97.1)	92.3% (77.6–97.6)	43.7% (28.7–86.1)	3.27 (0.94–11.3)	0.35 (0.18–0.67)	73.8% (57.9–86.1)
(2) Hematuria > 19.5 cells/µL	57.5% (39.2–74.5)	66.6% (29.9–92.5)	86.3% (70.6–94.3)	30% (18.9–44)	1.72 (0.65–4.55)	0.63 (0.34–1.17)	59.5% (43.2–74.3)
(3) Proteinuria > 2.9 g/day	48.5% (30.8–66.4)	88.8% (51.7–99.7)	94.1% (70.9–99.1)	32% (23.9–41.3)	4.36 (0.66–29.6)	0.58 (0.38–0.86)	57.1% (40.9–72.2)
(4) eGFR ≤ 81 mL/min	60.6% (42.1–77.1)	66.6% (29.9–92.5)	86.9% (71.7–94.5)	31.5% (19.7–46.3)	1.81 (0.69–4.76)	0.59 (0.31–1.1)	61.9% (45.6–76.4)
(5) C3 level ≤ 48 mg/dL	24.2% (11.1–42.2)	100% (66.3–100)	100% (63.1–100)	26.4% (22.8–30.3)	-	0.75 (0.62–0.92)	40.4% (25.6–56.7)
(6) C4 level ≤ 8 mg/dL	36.3% (20.4–54.8)	88.8% (51.7–99.7)	92.3% (64.1–98.7)	27.5% (21.2–35)	3.27 (0.48–21.9)	0.71 (0.51–1.01)	47.6% (32–63.5)
(7) ANA level > 5.9 U/mL	53.1% (34.7–70.9)	87.5% (47.3–99.6)	94.4% (72.5–99.1)	31.8% (22.8–42.3)	4.25 (0.66–27.3)	0.53 (0.34–0.84)	60% (43.3–75.1)
1 + 7	34.3% (18.5–53.2)	100% (63.1–100)	100% (71.5–100)	27.5% (22.8–32.8)	-	0.65 (0.51–0.84)	47.5% (31.5–63.8)
1 + 7 + 3	18.7% (7.2–36.4)	100% (63.1–100)	100% (54.1–100)	23.5% (20.6–51.6)	-	0.81 (0.68–0.96)	35% (20.6–51.6)
1 + 7 + 3 + 4	12.5% (3.51–28.9)	100% (63.1–100)	100% (39.7–100)	22.2% (20.04–24.5)	-	0.87 (0.76–0.99)	30% (16.5–46.5)
1 + 7 + 3 + 4 + 5	9.37% (1.97–25.02)	100% (63.1–100)	100% (29.2–100)	21.6% (19.7–23.5)	-	0.91 (0.81–1.01)	27.5% (14.6–43.8)
**(II) AI over 3**							
(1) usCD163 > 296.2 ng/mmol	75% (56.5–88.5)	80% (44.4–97.4)	92.3% (77.3–97.6)	50% (33.7–66.2)	3.75 (1.06–13.1)	0.31 (0.16–0.61)	76.2% (60.5–87.9)
(2) Hematuria > 19.5 cells/µL	59.3% (40.6–76.3)	70% (34.7–93.3)	86.3% (70.2–94.4)	35% (23.1–49.1)	1.98 (0.73–5.32)	0.58 (0.32–1.04)	61.9% (45.6–76.4)
(3) Proteinuria > 2.9 g/day	50% (31.8–68.1)	90% (55.5–99.7)	94.1% (70.7–99.1)	36% (27.3–45.7)	5 (0.75–33.1)	0.55 (0.37–0.83)	59.5% (43.2–74.3)
(4) eGFR ≤ 130 mL/min	100% (89.1–100)	20% (2.52–55.6)	80% (74.5–84.5)	100% (15.8–100)	1.25 (0.91–1.7)	-	80.9% (65.8–91.4)
(5) C3 level ≤ 48 mg/dL	25% (11.4–43.4)	100% (69.1–100)	100% (63.1–100)	29.4% (25.4–33.7)	-	0.75 (0.61–0.91)	42.8% (27.7–59.04)
(6) C4 level ≤ 8 mg/dL	37.5% (21.1–56.3)	90% (55.5–99.7)	92.3% (63.9–98.8)	31.03% (24.3–38.7)	3.75 (0.55–25.4)	0.69 (0.49–0.97)	50% (34.2–65.8)
(7) ANA level > 7.6 U/mL	32.2% (16.6–51.3)	100% (66.3–100)	100% (69.1–100)	30% (25.1–35.3)	-	0.67 (0.53–0.86)	47.5% (31.5–63.8)
1 + 3	46.8% (29.1–65.2)	90% (55.5–99.7)	93.7% (69.2–99)	34.6% (26.4–43.7)	4.68 (0.7–31.2)	0.59 (0.4–0.86)	57.1% (40.9–72.2)
1 + 3 + 7	6.45% (0.79–21.4)	100% (66.3–100)	100% (15.8–100)	23.6% (22.1–25.4)	-	0.93 (0.85–1.02)	27.5% (14.6–43.8)
1 + 3 + 7 + 5	3.22% (0.08–16.7)	100% (66.3–100)	100% (2.5–100)	23.1% (21.9–24.2)	-	0.96 (0.91–1.03)	25% (12.6–41.2)
1 + 3 + 7 + 5 + 2	3.22% (0.08–16.7)	100% (66.3–100)	100% (2.5–100)	23.1% (21.9–24.2)	-	0.96 (0.91–1.03)	25% (12.6–41.2)
**(III) AI of at least 9**							
(1) usCD163 > 296.2 ng/mmol	88.8% (65.2–98.6)	58.3% (36.6–77.9)	61.5% (49.2–72.5)	87.5% (64.4–96.4)	2.13 (1.29–3.52)	0.19 (0.05–0.73)	71.4% (55.4–84.2)
(2) Hematuria > 19.5 cells/µL	77.7% (52.3–93.6)	66.6% (44.6–84.3)	63.6% (48.5–76.4)	80% (61.7–90.8)	2.33 (1.26–4.32)	0.33 (0.13–0.82)	71.4% (55.4–84.2)
(3) Proteinuria > 1.4 g/day	83.3% (58.6–96.4)	45.8% (25.5–67.2)	53.6% (43.1–63.7)	78.5% (54.4–91.8)	1.54 (1.009–2.34)	0.36 (0.12–1.11)	61.9% (45.6–76.4)
(4) eGFR ≤ 100 mL/min	88.8% (65.2–98.6)	33.3% (15.6–55.3)	50% (41.9–58.1)	80% (49.1–94.3)	1.33 (0.96–1.84)	0.33 (0.08–1.38)	57.1% (40.9–72.2)
(5) C3 level ≤ 41.2 mg/dL	33.3% (13.3–59)	95.8% (78.8–99.9)	85.7% (44.1–97.8)	65.7% (57.7–72.8)	8 (1.05–60.7)	0.69 (0.49–0.97)	69.05% (52.9–82.3)
(6) C4 level ≤ 4.6 mg/dL	11.1% (1.37–34.7)	100% (85.7–100)	100% (15.8–100)	60% (56.02–63.8)	-	0.88 (0.75–1.04)	61.9% (45.6–76.4)
(7) ANA level > 9.6 U/mL	29.4% (10.3–55.9)	95.6% (78.05–99.9)	83.3% (39.1–97.5)	64.7% (57.1–71.6)	6.76 (0.86–52.7)	0.74 (0.53–1.01)	67.5% (50.8–81.4)
1 + 2	66.6% (40.9–86.6)	83.3% (62.6–95.2)	75% (53.6–88.6)	76.9% (62.8–86.7)	4 (1.54–10.3)	0.4 (0.2–0.78)	76.2% (60.5–87.9)
1 + 2 + 3	61.1% (35.7–82.7)	83.3% (62.6–95.2)	73.3% (51.1–87.8)	74.1% (60.9–83.9)	3.66 (1.39–9.64)	0.46 (0.25–0.85)	73.8% (57.9–86.1)
1 + 2 + 3 + 7	17.6% (3.8–43.4)	100% (85.1–100)	100% (29.2–100)	62.2% (56.8–67.2)	-	0.82 (0.66–1.02)	65% (48.3–79.3)
1 + 2 + 3 + 7 + 5	11.7% (1.45–36.4)	100% (85.2–100)	100% (15.8–100)	60.5% (56.3–64.5)	-	0.88 (0.74–1.05)	62.5% (45.8–77.2)

**Abbreviations:** LR+: positive likelihood ratio; LR-: negative likelihood ratio; eGFR: estimated glomerular filtration rate; ANA: antinuclear antibodies.

## Data Availability

The data that support the findings of this study are available from the corresponding author upon reasonable request. The data are not publicly available due to privacy or ethical restrictions.

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
