# Peer review of "Clinical Predictors of Underlying Histologic Activity in Patients with Lupus Nephritis: A Focus on Urinary Soluble CD163"

_jcm, 2025, doi:10.3390/jcm14176162_

Round 1

Reviewer 1 Report

Comments and Suggestions for Authors

This a very well designed study exploring the value of sCD163 in predicting activity in lupus nephritis. Fig. and Tab. are fine, in Tab.1 the separation of the 2 columns could more clear.

The data are confirming earlier results, unfortunately sCD163 level is not superior to proteinuria the actual clinical standard marker. In the discussion, the authors describe this limitation to discriminate between proteinuria and inflammation. Cases with proteinuria caused by chronicity may be helpful here for this differentiation, and I cannot see whether those are in the cohort. Another option could be, as discussed by the authors, to extract patients with low proteinuria and high activity score.

One weakness of the interpretation of the data ist, that the most important urine biomarker analysis of lupus nephritis is ignored by the authors. They should include the data provided by Andrea Fava et al. in the evaluating of their data. There is sCD163 only one the identified biomarkers, but it is better seen in context of others. In this paper there is focus on one biomarker only.

In the basic despription, disease duration and manifestation duration are missing, which are important for the interpretation of the results. 

Minor points:
L41 LN is not the leading cause of morbidity and mortality in SLE.

Author Response

Dear Journal of Clinical Medicine Editorial team,

On behalf of the co-authors, I want to thank you for the opportunity of incorporating reviewer comments made in relation to our manuscript entitled “Clinical predictors of underlying histologic activity in patients with lupus nephritis: a focus on urinary soluble CD163”. We hope to have addressed all the comments and suggestions and believe that it has made our report clearer and more meaningful for publication.

Sincerely,

Bogdan Obrisca

Reviewer 1

Comment 1: This a very well designed study exploring the value of sCD163 in predicting activity in lupus nephritis. Fig. and Tab. are fine, in Tab.1 the separation of the 2 columns could more clear.

Response 1: We are thankful for the appreciation and thorough review of our manuscript that provided us essential guidance to definitely improve the quality of the manuscript. We have separated as per your suggestions Tabel 1 into two tables to be more clear and in order to incorporate the additional information requested by other reviewers.

Comment 2: The data are confirming earlier results, unfortunately sCD163 level is not superior to proteinuria the actual clinical standard marker. In the discussion, the authors describe this limitation to discriminate between proteinuria and inflammation. Cases with proteinuria caused by chronicity may be helpful here for this differentiation, and I cannot see whether those are in the cohort. Another option could be, as discussed by the authors, to extract patients with low proteinuria and high activity score.

Response 2: Thank you for this important observation. Indeed, in our cohort the usCD163 level had slightly better performances to discriminate active inflammation, but was not statistically superior to proteinuria. One reason may be the limited number of patients. The ability to discriminate residual proteinuria versus active inflammation remains tricky as proteinuria is responsible for up to 40-60% of the variation of usCD163 level, but not 100%. Thus, a contribution of both proteinuria and active inflammation is reflected in the urinary levels of usCD163, and whether normalization of usCD163 levels to urinary albumin will provide a better imagine of the active glomerular inflammation remains to be answered. In order to further emphasize your suggestion and the future utility of usCD163, we included Figure 4 that stratified usCD163 level, activity and chronicity index according to baseline proteinuria. In addition, we further included in the result section several descriptions of individual patients to highlight these aspects.

Comment 3: One weakness of the interpretation of the data is, that the most important urine biomarker analysis of lupus nephritis is ignored by the authors. They should include the data provided by Andrea Fava et al. in the evaluating of their data. There is sCD163 only one the identified biomarkers, but it is better seen in context of others. In this paper there is focus on one biomarker only.

Response 3: Thank you for the suggestion. We agree that other biomarkers (and therefore a combination of those) would reflect more reliably the underlying immune and pathological landscape of the patients with lupus nephritis. At this point we only had the opportunity to evaluate usCD163, but in the future, as more funding will be available, we will expand the evaluation of other potential serum or urine biomarkers (as samples are available and stored adequately in our department). We have updated the limitations section of the manuscript and introduce the reference of Andreea Fava et al (Fava A, Buyon J, Magder L, Hodgin J, Rosenberg A, Demeke DS, Rao DA, Arazi A, Celia AI, Putterman C, Anolik JH, Barnas J, Dall'Era M, Wofsy D, Furie R, Kamen D, Kalunian K, James JA, Guthridge J, Atta MG, Monroy Trujillo J, Fine D, Clancy R, Belmont HM, Izmirly P, Apruzzese W, Goldman D, Berthier CC, Hoover P, Hacohen N, Raychaudhuri S, Davidson A, Diamond B; Accelerating Medicines Partnership in RA/SLE network; Petri M. Urine proteomic signatures of histological class, activity, chronicity, and treatment response in lupus nephritis. JCI Insight. 2024 Jan 23;9(2):e172569).

Comment 4: In the basic description, disease duration and manifestation duration are missing, which are important for the interpretation of the results.

Response 4: Thank you for the suggestion. We have added the disease duration in the result section and Table 1.

 Minor points:

Comment 5: L41 LN is not the leading cause of morbidity and mortality in SLE.

Response 5: We have rephrased the sentence.

Reviewer 2 Report

Comments and Suggestions for Authors

The study by Obrigcā et al. investigates the utility of urinary soluble CD163 (usCD163) as a biomarker for histologic activity in lupus nephritis (LN). The research is well-designed and addresses an important clinical gap in non-invasive monitoring of LN. However, there are several areas where the manuscript could be improved for clarity, statistical robustness, and clinical applicability.

Major Comments:

    • Consider collaborating with other centers to increase the sample size or explicitly state the need for multi-center validation in future research.
    • Perform multivariate analyses to adjust for proteinuria and other confounders to determine if usCD163 provides independent predictive value. Additionally, discuss the biological plausibility of usCD163 as a marker of inflammation versus a byproduct of proteinuria.
    • Expand the longitudinal cohort or provide a more detailed discussion of the limitations of the current data. Highlight the need for larger longitudinal studies to assess usCD163's role in monitoring treatment response.
    • Validate this cutoff in an independent cohort or discuss plans for future validation. Additionally, explore whether the cutoff varies by LN class or other clinical factors.
    • Emphasize this point in the abstract and discussion, as it tempers the conclusion that usCD163 is a superior biomarker. Discuss the implications for clinical practice, such as whether usCD163 should be used alongside or instead of proteinuria.

Minor Comments:

    • Provide details on sample storage duration and stability of usCD163 under these conditions.
    • Ensure all figures are included or clarify their absence (e.g., if they are supplemental).
    • Discuss how these factors were controlled for or their potential impact on the results.
    • Provide a hypothetical algorithm or scenario where usCD163 testing could guide treatment decisions, such as when to perform a repeat biopsy or adjust immunosuppression.

Author Response

Dear Journal of Clinical Medicine Editorial team,

On behalf of the co-authors, I want to thank you for the opportunity of incorporating reviewer comments made in relation to our manuscript entitled “Clinical predictors of underlying histologic activity in patients with lupus nephritis: a focus on urinary soluble CD163”. We hope to have addressed all the comments and suggestions and believe that it has made our report clearer and more meaningful for publication.

Sincerely,

Bogdan Obrisca

Reviewer 2

Comment 1: The study by Obriscā et al. investigates the utility of urinary soluble CD163 (usCD163) as a biomarker for histologic activity in lupus nephritis (LN). The research is well-designed and addresses an important clinical gap in non-invasive monitoring of LN. However, there are several areas where the manuscript could be improved for clarity, statistical robustness, and clinical applicability.

Response 1: We are thankful for the appreciation and thorough review of our manuscript that provided us essential guidance to definitely improve the quality of the manuscript. We hope we addressed all the concerns that you raised.

Major Comments:

Comment 2: Consider collaborating with other centers to increase the sample size or explicitly state the need for multi-center validation in future research.

Response 2: Thank you for the suggestion. We acknowledge this limitation and updated the study limitations section accordingly.

Comment 3: Perform multivariate analyses to adjust for proteinuria and other confounders to determine if usCD163 provides independent predictive value. Additionally, discuss the biological plausibility of usCD163 as a marker of inflammation versus a byproduct of proteinuria.

Response 3: Thank you for the suggestion. We added the multivariate analysis in Table S3 (not to overcrowd the article) and introduced the findings in the result section. We have tried to highlight in the discussion section the notion that proteinuria accounts for about 40-50% of the variability in usCD163 and whether normalization to urinary albumin will be more reliable, but this will be subject to future research. Nonetheless, to further stratify the role of usCD163 we introduce in the result section Figure 4 that presents the usCD163 level, activity and chronicity index stratified by baseline proteinuria. In addition, also in the result section we detailed several individual cases from the study cohort with different proteinuria level and different histological findings.

Comment 4: Expand the longitudinal cohort or provide a more detailed discussion of the limitations of the current data. Highlight the need for larger longitudinal studies to assess usCD163's role in monitoring treatment response.

Response 4: We have updated the study limitations section to further highlight your suggestions and the need for larger longitudinal studies.

Comment 5: Validate this cutoff in an independent cohort or discuss plans for future validation. Additionally, explore whether the cutoff varies by LN class or other clinical factors.

Emphasize this point in the abstract and discussion, as it tempers the conclusion that usCD163 is a superior biomarker. Discuss the implications for clinical practice, such as whether usCD163 should be used alongside or instead of proteinuria.

 Response 5: Similar to previous suggestions, we have updated the limitation section according to your suggestions. In addition, we have tried to explore previously whether cutoff varies by LN class, but the limited number of patients in this cohort prevented us from drawing any stronger conclusions. Thus, we feel that including further stratification for this cohort will create ambiguity for our results. In addition, as previously mentioned, in the result section we introduced several examples of patients with different baseline proteinuria and histological features to further highlight the role of usCD163 in different clinical scenarios.

Minor Comments:

Comment 6: Provide details on sample storage duration and stability of usCD163 under these conditions.

Response 6: We have introduced additional details on sample collection, storage and stability of usCD163 in both methods section and discussion section.

Comment 7: Ensure all figures are included or clarify their absence (e.g., if they are supplemental).

Response 7: We have revised all figures, included additional ones as per suggestions in order to fully incorporate all the essential data in the manuscript.

Comment 8: Discuss how these factors were controlled for or their potential impact on the results.

Response 8: We are not sure of what factors you are referring to and would be grateful for further guidance. In case of sample handling and processing, we have updated the methods and discussion section accordingly.

Comment 9: Provide a hypothetical algorithm or scenario where usCD163 testing could guide treatment decisions, such as when to perform a repeat biopsy or adjust immunosuppression.

Response 9: Thank you for the suggestion. Although at this point of knowledge there is insufficient data to provide firm recommendations on optimal monitoring interval of usCD163, we have introduced in the discussion section a paragraph suggesting a potential monitoring algorithm for usCD163.

Reviewer 3 Report

Comments and Suggestions for Authors

At the beginning, there is a slightly higher percentage of overlap in the text that was confirmed by iThenticate, so the method of interpreting references needs to be reviewed in more detail!

- is the number of patients in whom the study is acceptable, but the study design is questionable considering the different classes of lymph node nuclearity.

- Introduce ethical approval of the institution where the study was conducted into the methodology!

- Divide the results of the findings according to the class of lymph node nuclearity and according to the therapy!

Also introduce the lipid profile in Table 1!

- Add anti-DS DNA as well as anti-Sm antibodies to immunology.

- What therapy was used! cyclophosphamide, MMF and others.

The discussion is excellently written and the conclusion is drawn from the results!

Author Response

Dear Journal of Clinical Medicine Editorial team,

On behalf of the co-authors, I want to thank you for the opportunity of incorporating reviewer comments made in relation to our manuscript entitled “Clinical predictors of underlying histologic activity in patients with lupus nephritis: a focus on urinary soluble CD163”. We hope to have addressed all the comments and suggestions and believe that it has made our report clearer and more meaningful for publication.

Sincerely,

Bogdan Obrisca

Reviewer 3

Comment 1: At the beginning, there is a slightly higher percentage of overlap in the text that was confirmed by iThenticate, so the method of interpreting references needs to be reviewed in more detail!

Response 1: Thank you for the careful review of our manuscript. We have revised our initial phrases from the introduction.

Comment 2: - is the number of patients in whom the study is acceptable, but the study design is questionable considering the different classes of lymph node nuclearity.

Response 2: Thank you for the suggestion. Our study design is indeed a cross-sectional one that we acknowledged in the study limitation section that was updated to fully clarify this aspect. In addition, we have revised the biopsies of these patients to identify the description of tertiary lymphoid organs and introduce these findings in the result section (6 patients had tertiary lymphoid organs). However, our pathologist does not report the different stages of tertiary lymphoid organ development and we could not fully address your suggestion.  

Comment 3: - Introduce ethical approval of the institution where the study was conducted into the methodology!

Response 3: The ethical approval of the institution is already introduced both in the methods section and at the end of the manuscript prior to references according to journal guidelines.

Comment 4: - Divide the results of the findings according to the class of lymph node nuclearity and according to the therapy!

Response 4: We have included the percentage of patients with presence of tertiary lymphoid organs and the corresponding usCD163 level (figure 5 in the result section). We have included data regarding the therapy of these patients. However, given that all patients were treated an analysis according to therapy in our study (given the limited number of patients) was not possible. However, we will take this into account in the future as an expansion of this cohort will be available.

Comment 5: Also introduce the lipid profile in Table 1!

Response 5: Thank you for the suggestion. We have added the lipid profile in Table 1.

Comment 6: - Add anti-DS DNA as well as anti-Sm antibodies to immunology.

Response 6: Thank you for the suggestion. We have added the data regarding the positivity of anti-ds DNA and anti-Sm antibodies in Table 1.

Comment 7: - What therapy was used! cyclophosphamide, MMF and others.

Response 7: We have added the data regarding type of induction and maintenance treatment of these patients in Table 1.

Comment 8: The discussion is excellently written and the conclusion is drawn from the results!

Response 8: We are thankful for the sincere appreciation and constructive comments.

Reviewer 4 Report

Comments and Suggestions for Authors

Dear Authors!

Thank you for the opportunitty to review your manuscript

SLE with kidney involvememnt is a severe disases with restriction of life expactancy

SLE is difficult to measure its activity despite the numerous clinical, laboratorial and morphological markers.

New easy-performed and effective non-invasive biomarkers of the LN activity is neede.

Authors assessed the role of usCD163 as biomarkers of the LN in SLE patients and the results are intersting and promising

The study is actual and the introduction underline the actuality of the study

The Methods described in detaiks

The results are clear and only a few clarifications required

Author provided the sensitivity and specificity analysis with cutt-offs calculation with AUC-ROC analysis and provided the diagnostic models

The diascussion contain the relevant literature and the Authors compared their results with the prevuoisly published data

The discussion has the expand Limitations subsection

The conlusion supports the study results

During the review I have severall minor suggestions

1) Statistics - please add the statement about the assessment of the ditribution of the quantitative data and priniciples for choosing the parametric and non-parametric tests

2) Table 2 miss some absolute number, only percentage, please add the abs numbers

3) Table 3 consists of the three parts and every part lasts by the combinations of the predictors. Can I ask to explain the selection of the factors? May be beteer to use regression analysis to avoid the manual combination? and decrease the number of diagnostic models?

In common the manuscript is well-written and well-organized

Good Luck!

Reviewer

Author Response

Dear Journal of Clinical Medicine Editorial team,

On behalf of the co-authors, I want to thank you for the opportunity of incorporating reviewer comments made in relation to our manuscript entitled “Clinical predictors of underlying histologic activity in patients with lupus nephritis: a focus on urinary soluble CD163”. We hope to have addressed all the comments and suggestions and believe that it has made our report clearer and more meaningful for publication.

Sincerely,

Bogdan Obrisca

Reviewer 4

Dear Authors!

Comment 1: Thank you for the opportunity to review your manuscript

SLE with kidney involvement is a severe disease with restriction of life expectancy

SLE is difficult to measure its activity despite the numerous clinical, laboratorial and morphological markers.

New easy-performed and effective non-invasive biomarkers of the LN activity is needed.

Authors assessed the role of usCD163 as biomarkers of the LN in SLE patients and the results are interesting and promising

The study is actual and the introduction underline the actuality of the study

The Methods described in details

The results are clear and only a few clarifications required

Author provided the sensitivity and specificity analysis with cutt-offs calculation with AUC-ROC analysis and provided the diagnostic models

The discussion contain the relevant literature and the Authors compared their results with the previously published data

The discussion has the expand Limitations subsection

The conclusion supports the study results

Response 1: We are thankful for the appreciation and thorough review of our manuscript that provided us essential guidance to definitely improve the quality of the manuscript.

During the review I have several minor suggestions

Comment 2: 1) Statistics - please add the statement about the assessment of the distribution of the quantitative data and principles for choosing the parametric and non-parametric tests

Response 2: Thank you for the suggestion. We have added the statement about the assessment of distribution variables. In addition, given the fact the usCD163 and the majority of other variables were non-normally distributed, for the correlation of variables we used Spearman’s rank correlation test, as stated. For comparison between groups we used Mann-Whitney test. All the information was updated in the methods section.

Comment 3: 2) Table 2 miss some absolute number, only percentage, please add the abs numbers

Response 3: Thank you for the suggestion. We have updated the table.

Comment 4: 3) Table 3 consists of the three parts and every part lasts by the combinations of the predictors. Can I ask to explain the selection of the factors? May be better to use regression analysis to avoid the manual combination? and decrease the number of diagnostic models?

Response 4: Thank you for this important observation. Indeed, we stated in the methods section the following sentence: “In order to evaluate whether combining clinical parameters will improve the performance characteristics, several models were developed by progressively combining the first five variables with the highest AUC for a given activity index threshold. We chose this method in order to see if combining usCD163 with the classical measures of LN activity, as used in common clinical practice, improves diagnostic sensitivity. In addition, we introduced in the supplementary files and result section a univariate and multivariate logistic regression analysis regarding predictive factors for an elevated usCD163 using the predefined cutoff by AUC analysis.

In common the manuscript is well-written and well-organized

Round 2

Reviewer 1 Report

Comments and Suggestions for Authors

Now the data are better understandable, I am still missing the time to treatment.

Author Response

Dear Journal of Clinical Medicine Editorial team,

On behalf of the co-authors, I want to thank you for the opportunity of incorporating reviewer comments made in relation to our manuscript entitled “Clinical predictors of underlying histologic activity in patients with lupus nephritis: a focus on urinary soluble CD163”. We hope to have addressed all the comments and suggestions and believe that it has made our report clearer and more meaningful for publication.

Sincerely,

Bogdan Obrisca

Comment 1: Now the data are better understandable, I am still missing the time to treatment.

Response 1: Thank you for your suggestion. Given the variability in the time of biopsy (either for initial diagnosis, during maintenance or for cause), we have entered in addition to time from SLE diagnosis to biopsy, the time from induction therapy start to biopsy (months) as to better reflect our assessment of usCD163 in relation to histology.

Reviewer 2 Report

Comments and Suggestions for Authors

The authors have successfully addressed all my comments

Author Response

Dear Journal of Clinical Medicine Editorial team,

On behalf of the co-authors, I want to thank you for the opportunity of incorporating reviewer comments made in relation to our manuscript entitled “Clinical predictors of underlying histologic activity in patients with lupus nephritis: a focus on urinary soluble CD163”. We hope to have addressed all the comments and suggestions and believe that it has made our report clearer and more meaningful for publication.

Sincerely,

Bogdan Obrisca

Comment 1: The authors have successfully addressed all my comments.

Response 1: We are thankful for the sincere appreciation and guidance throughout the revision process.